# Experimental Research on Microwave Ignition and Combustion Characteristics of ADN-Based Liquid Propellant

**DOI:** 10.3390/mi13040510

**Published:** 2022-03-25

**Authors:** Jiannan Shen, Yusong Yu, Xuhui Liu, Jie Cao

**Affiliations:** 1Hydrogen Energy and Space Propulsion Laboratory, School of Mechanical, Electronic and Control Engineering, Beijing Jiaotong University, Beijing 100044, China; 20126054@bjtu.edu.cn; 2Beijing Institute of Control Engineering, Beijing 100190, China; xhliu99@163.com; 3China North Engine Research Institute, Tianjin 300400, China; jackcao99@163.com

**Keywords:** microwave ignition system, ammonium dinitramide, microwave discharge, flame jet length

## Abstract

Microwave ignition has attracted much attention due to its advantages of reliable ignition, large ignition area and cold-start capability. In this paper, the experimental method is used to explore the ignition ability of the microwave device to ADN-based liquid propellant. Additionally, we discuss the influence of the inlet power and rate of propellant injected into the ignition system on the height of the combustion jet and the combustion temperature. In the experiment, a microwave-assisted ignition system was established based on a special microwave resonant cavity. The liquid propellant and working gas were sprayed into the resonator cavity through the hollow straight tube beneath the resonant cavity. The test results show that the device can ignite the propellant under the condition of 800 W input power, which proves the feasibility of the microwave ignition device for ADN-based liquid propellant. Microwave power has some influence on the flame spray height at the initial stage of combustion. The spray height at 2000 W is increased by 55.7% in comparison to 1000 W. In the stable combustion stage, the input power has a very significant increase in the average temperature of the flame, which is increased by 25%. The combustion is relatively better when the propellant flow rate is 30 mL/min, and the height of the flame spray increases by 25.2%. The increase in throughput did not have a significant impact on the flame temperature.

## 1. Introduction

Due to the operational requirements and environmental dangers of traditional propellants, rapidly developing aerospace technologies have put higher demands on propellants. As a new type of propellant with high performance and low toxicity [1], ADN (ammonium dinitramide)-based propellant represents, to some extent, a new research direction and a new trend in space chemical propulsion technology [2,3].

Flight safety and the durability of propulsion systems in aerospace engineering are closely connected to combustion control. Current ADN-based liquid thrusters primarily use preheating catalysis to obtain the ignition of ADN-based propellants [4,5,6,7]. Limited by the catalyst itself, the catalytic ignition method also hinders, to some extent, the development of the green propellants sector. Therefore, it is necessary to find a new ignition method that can ignite in a stable manner. Among them, electric ignition, microwave ignition, laser radiation, laser-induced breakdown and other ignition methods have been widely concerned.

The plasma affects combustion mainly via three different pathways: thermal, kinetic, and transport, and the three different enhancement pathways often couple together and improve the combustion. Microwave plasma belongs to non-equilibrium plasma, has higher electron temperature and is more kinetically active due to the rapid production of active radicals and excited species via electron impact dissociation, excitation, and subsequent energy relaxation [8,9,10]. Microwave plasma is widely used in material synthesis [11], organic surface modification [12] and biomedicine [13]. The plasma generated by microwaves is rich in charged particles, neutral particles in high energy states and highly reactive free radicals. It is theoretically possible to utilize this feature to ignite or support the combustion of all fuels. Microwave ignition devices have already been explored in vehicles and have made great strides [14,15,16,17]. After experiments, microwave plasma-assisted ignition has been shown to accelerate the combustion process based on the principle of generating non-equilibrium of chemically active particles. In reference [18], the flame enhancement is believed to be caused by a non-thermal chemical kinetic enhancement from energy deposition to free electrons in the flame front and induced flame from excitation of plasma instability. It has excellent research and practical value for ultra-lean flame, high-velocity flux, low-temperature and low-pressure conditions [8,19]. Constant volume combustion tests also demonstrate the ability of microwave-assisted combustion to extend the lean mass limit [10,11,12,13,14,15,16,17,18,19,20,21]. Compared to conventional spark plugs, microwave-assisted combustion shortens the combustion duration and display a higher pressure peak and heat release rate in the cylinder. The flame speed is 20% faster than a traditional spark plug.

In this work, an ignition test device has been set up and the ignition of ADN-based propellant has been carried out on the basis of the experimental microwave ignition platform. The influence of input power on microwave ignition under atmospheric conditions is previously analyzed, and the combustion conditions of various throughput of the propellant under the same input power are tested. However, this paper does not examine in detail the impact of working gas flow changes on the ignition and combustion of the propellant.

## 2. Experimental Setup and Program

### 2.1. Experimental Setup

The whole device (Figure 1 and Figure 2) consists of microwave generating system, cooling system, ignition system and measuring system. The cooling system consists of a cooling water tank and a water pump, which are responsible for the cooling of the device and prevent shutdown due to overheating. The microwave generation system consists of a microwave power supply (powering the entire device), a magnetron (generating microwaves), an automatic microwave tuning detection device (protecting the magnetron, matching circuit impedance) and a waveguide assembly (transmitting the generated microwaves to the ignition system), responsible for the generation of fixed frequency microwaves. The ignition system is composed of a waveguide connector, a customized aluminum resonant cavity, a peristaltic pump and a nozzle (the diameter of the nozzle is 0.1 mm), an intake and exhaust system, and an inner conductor of a tungsten needle. It is the place where the microwave-ionized working gas ignites the propellant. The size of the nozzle used in the experiment is 0.1 mm.

The measuring system consists of a thermocouple and a high-speed camera, which are used to measure the average temperature of the flame and photograph the height of the flame jet. Measuring equipment used in the experiment are high-speed camera and temperature sensor. The high-speed camera is Photron nova S9. It has 1 million pixels and 800 fps. Temperature sensor used in this study is Type K armored thermocouple, which has measure between 0 and 1500 °C. In the experiments, we set the thermocouple at 10 cm from the burner exit.

A microwave power supply powers the generation system, and the device can supply up to 3 kW of power to the magnetron. The microwave energy generated by the magnetron at 2.45 GHz is sent to a tuning detection device, which automatically tunes to maximize the power delivered to the waveguide assembly after measuring the forward and reflected power. Relying on the dummy load to absorb the reflected power avoids the reflected power from harming the magnetron. During this period, the cooling water is supplied by the water pump to realize the cooling of the whole device.

The microwave energy leaving the microwave generating device enters the waveguide connector for further transfer to the resonant cavity. The custom aluminum resonator takes the form of a transitional reduction in height to increase the electric field strength without changing the transmission wavelength and frequency of the waveguide. A connection device between the propellant and the intake and exhaust is set at the place where the electric field of the resonant cavity is the strongest, and the propellant is sprayed into the ignition system by means of a peristaltic pump.

### 2.2. Microwave Resonator Designs

In order to realize the ignition of ADN-based propellants, a sufficient electric field strength (at least equal to 3 × 10^6^ V/m) needs to be formed in the ignition device, so a microwave resonant cavity capable of reaching a specific electric field strength is necessary. In this paper, an aluminum resonator is fabricated based on the BJ26 rectangular metal waveguide. It has a short-circuit end and a half-height modification is designed compared with the waveguide. Table 1 shows the dimensions of the resonant cavity, and the shape of the resonator is shown in Figure 3.

The ignition area should be placed where the point field of the resonator is strongest. To explore where the resonator’s electric field is strongest, the resonator is modeled in Comsol Multiphysics. In this study, the RF radio frequency module is used for calculation in the steady-state frequency domain, a similar method has been used in reference [22]. The electric field inside the resonator cavity is analyzed based on the finite element method. The applied mathematical model is Maxwell equations, the boundary condition is set as perfect-E conductor, the interior of the cavity is set as air, and the cavity material is set as aluminum. Since the cavity is made of rectangular waves, the input port type is selected as rectangular, and the main mode inside the cavity is the TE_10_ mode. The working gas and propellant supply pipelines are arranged at the position of the strongest electric field. The pipeline can be regarded as a quartz tube with air inside. Figure 3 shows the meshing diagram of the resonant cavity model with a quartz tube. The reaction area has been refined, and the overall mesh number is 444,239, meshed by Comsol software (see Figure 4). Based on this model, the inner conductor insertion depth was explored.

Figure 5 shows the electric field distribution inside the half-height resonant cavity. It can be seen that the strongest electric field is located about 50 mm away from the short-circuit end.

The inner conductor will distort the electric field in the resonant cavity, destroying the original electric field distribution, while greatly increasing the electric field intensity around the inner conductor. The ionization of air can be achieved at 3 × 10^6^ V/m under atmospheric conditions (the solid blue line in Figure 6). According to the simulation results, it can be found that with the increase in the insertion depth, the electric field intensity at the tip of the inner conductor also increases continuously. The insertion depth of the inner conductor was fixed at 11 mm.

### 2.3. Experimental Program

In order to discuss the effects of input power and propellant flow rate of microwave ignition device on combustion characteristics, experimental studies under 9 working conditions were carried out. Before the experiment, firstly we need to ensure the normal circulation of cooling water in the device, and secondly ensure the stable supply of working gas (10 L/min). Microwave ignition is realized by adjusting the power and working gas to determine the feasibility of microwave ignition. Then, the exploration experiment is carried out, which is divided into two parts. One part of the experiment set the input power to 1000 W, 1250 W, 1500 W, 1750 W and 2000 W, respectively, under the condition of ensuring the propellant flow rate of 35 mL/min, to explore the influence of input power on the length of the flame jet and the flame temperature. In another part of the experiment, the input power was stabilized at 1500 W, and the propellant flow was set to 25 mL/min, 30 mL/min, 35 mL/min, 40 mL/min, and 45 mL/min, respectively, to explore the effect of propellant flow on the flame jet length and flame temperature. In the exploration experiment, the working gas flow was kept constant, the propellant flow was controlled by the peristaltic pump, and the input power of the microwave source was controlled by the computer software MX024.

It is worth noting that some white particles were found in the quartz tube after the test, which was caused by the incomplete combustion of the ADN-based liquid propellant, and part of the propellant was ejected out of the combustion area along with the working gas.

## 3. Results and Discussion

Theoretically, the ignition capability of the microwave device is only related to the incident power and the flow rate of working gas. Its combustion condition is also related to the flow rate of the propellant. Based on the constant working gas flow rate (10 L/min), the influence of microwave power and propellant flow rate on the combustion effect is explored.

### 3.1. Effects of Microwave Power on the Ignition and Combustion

Preliminary experiments show that when the input power is low, no visible plasma torch is generated in the cavity. As the input power increases, the generated discharge effect will ionize the working gas and generate a plasma torch. Based on the constant incident flow of propellant, the microwave power is set to 1000 W, 1250 W, 1500 W, 1750 W, 2000 W, and the influence of microwave power on the combustion effect of the flame is explored. The consistency of the plasma jet direction was ensured by keeping the position of the inner conductor in the combustion system unchanged throughout the test.

The working gas also provides a certain thrust for the plasma torch, so that the flame can take on the shape of a torch. Figure 7a shows the change of the flame shape of the plasma torch under different microwave powers during the initial combustion. It can be found that as the power increases, the height of the flame jet also increases. The highest increase in the height of the flame jet is about 55.7% (Figure 8), which shows that the increase in input power causes more powerful discharge field strength, forcing more gas to be broken down to form a plasma, thereby forming a larger-scale plasma torch. After the combustion is stable, the heights of the flames with these powers are not very different, and no longer as obvious as in the initial stage of ignition. A similar discovery was mentioned in reference [17], wherein microwave duration would not change the total heat release.

Figure 9 shows the effect of only changing input power on the average temperature of flame under the same propellant flow rate. It can be found that the increase in input power makes the average temperature of the propellant combustion flame increase significantly, which can be attributed to the increase in the microwave feed energy. It leads to more intense discharge, and the plasma torch itself has higher energy content, which is beneficial to the combustion of ADN-based liquid propellant. Compared with the case of 1000 W, the average temperature of the flame at 2000 W is increased by 25%.

### 3.2. Effects of Propellant Flow on the Ignition and Combustion

Under the condition that the microwave input power is kept unchanged at 1500 W, the influence of the propellant on the combustion characteristics is explored by controlling the flow rate (25–45 mL/min) of the propellant injected into the ignition device. We can conclude that in the case of 30 mL/min, the height of the flame reaching a peak, while the 35 mL/min decreased compared with 30 mL/min, and then slowly recovered (Figure 10). The maximum flame jet height difference is about 25.2%. The height of the flame jet also tends to be the same after the combustion is stable (Figure 11).

In order to learn more about the effect of the increase in propellant on the height of the flame jet after 45 mL/min, a subsequent experiment with a larger flow rate was carried out. Through the test, it was found that the increase in the flow rate caused a certain increase in the height of the flame jet. However, the flame shape became slender, and the propellant residue in the tube increased significantly. This shows to a certain extent that the combustion completeness of microwave ignition has a certain relationship with the propellant flow rate and atomization effect. In the case of the atomization aperture (0.1 mm), an excessive flow rate will make the propellant and the plasma torch. It is difficult to achieve sufficient contact, resulting in incomplete combustion.

It can be seen from Figure 12 that the increase in the propellant flow rate has no significant effect on the flame temperature. The temperature difference caused by the propellant flow rate is within 30 K. The flame temperature is the highest at 30 mL/min, and then there is a large drop. The phenomenon may be caused by the ability of the device itself, and too much propellant may not help the combustion. The overall effect of the propellant flow on the average temperature of the flame is relatively small, due to the appearance of large amounts of residues, which means the incomplete combustion.

## 4. Conclusions

The ADN-based liquid propellant was successfully ignited by the microwave device constructed. Compared with the traditional catalytic ignition, the microwave ignition achieved the purpose of cold start and could be ignited at multiple points. The whole experiment tested the effect of different microwave power and ADN-based liquid propellant flow on flame combustion, and reached the following conclusions:

(1) Keeping the propellant flow constant, the plasma flame jet height increased significantly with the increase in microwave power in the early stage of combustion, with the highest increase of 55.7% for 2000 W as compared to 1000 W. The temperature also increased significantly with the increase in microwave power, with a maximum increase of 25% increased for 2000 W as compared to 1000 W.

(2) In the case of 1500 W input power, when the propellant flow rate is kept at 30 mL/min, the propellant can be fully contacted and burned with the plasma flame in the discharge chamber. Excessive propellant can increase the height of the flame jet, but at the same time, it will make the combustion incomplete, resulting in the problem of a large amount of propellant residue.

## Figures and Tables

**Figure 1 micromachines-13-00510-f001:**
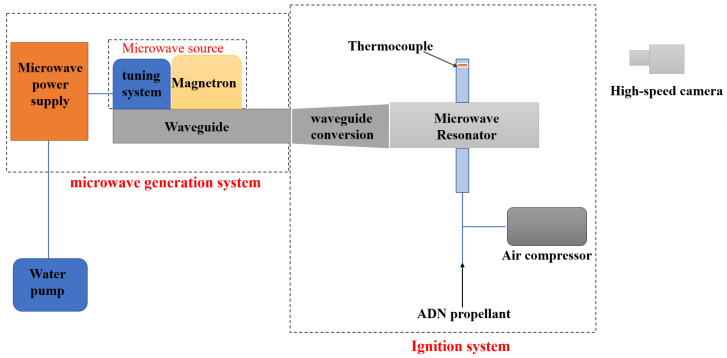
System diagram of microwave ignition test device.

**Figure 2 micromachines-13-00510-f002:**
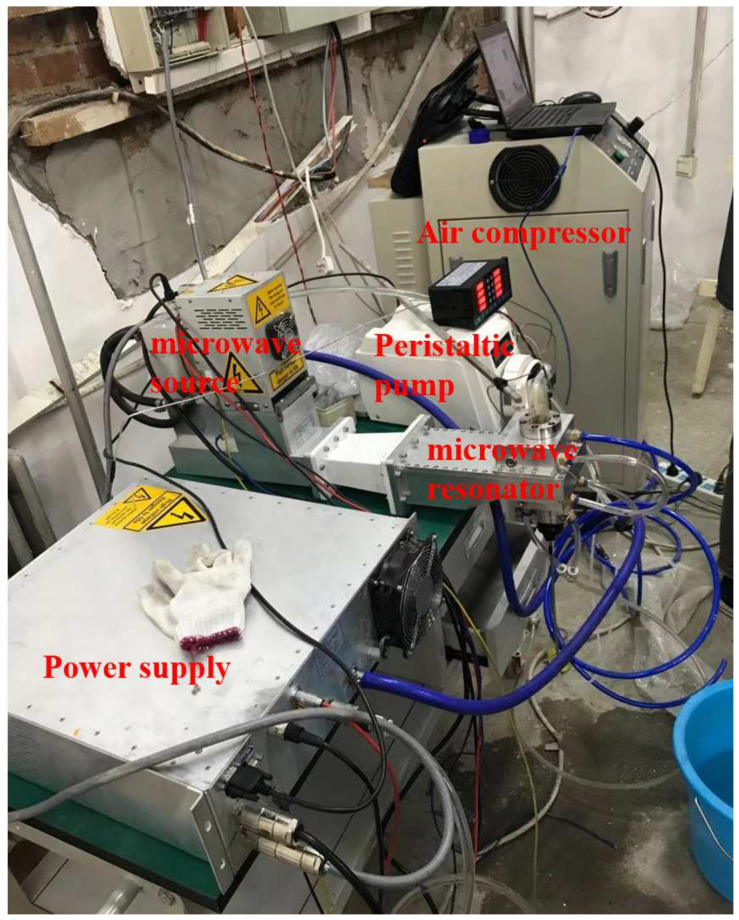
Real picture of microwave ignition device.

**Figure 3 micromachines-13-00510-f003:**
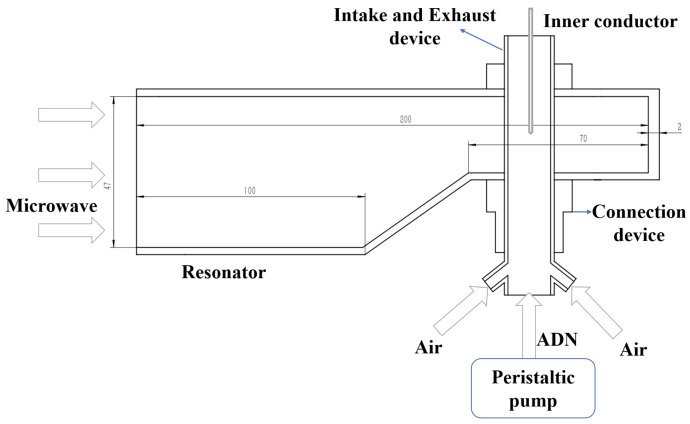
Schematic diagram of the microwave resonator.

**Figure 4 micromachines-13-00510-f004:**
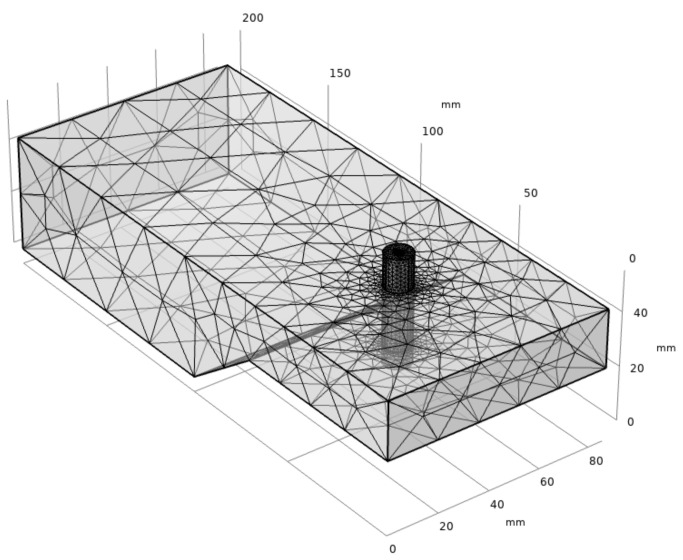
Meshing diagram of the resonator model.

**Figure 5 micromachines-13-00510-f005:**
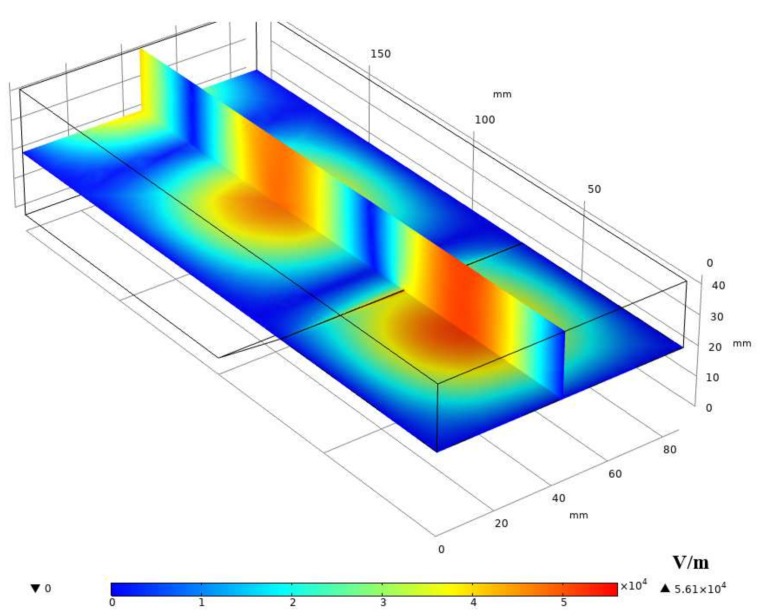
Electric field distribution in the cavity.

**Figure 6 micromachines-13-00510-f006:**
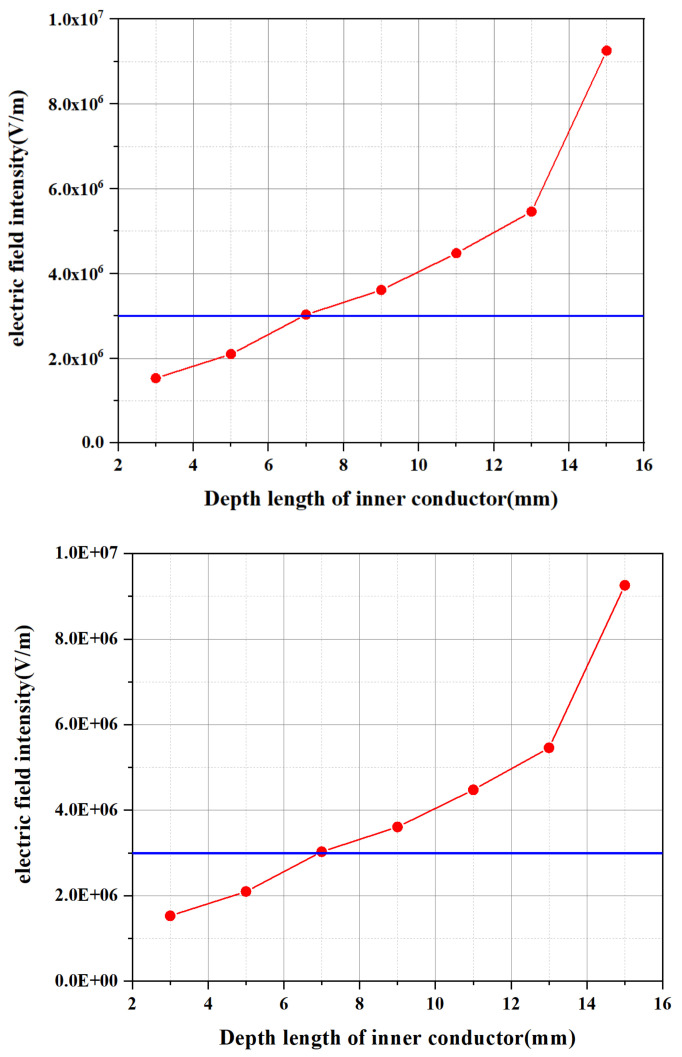
Electric field strength around the inner conductor (the blue line represents the electric field strength required for air to be ionized).

**Figure 7 micromachines-13-00510-f007:**
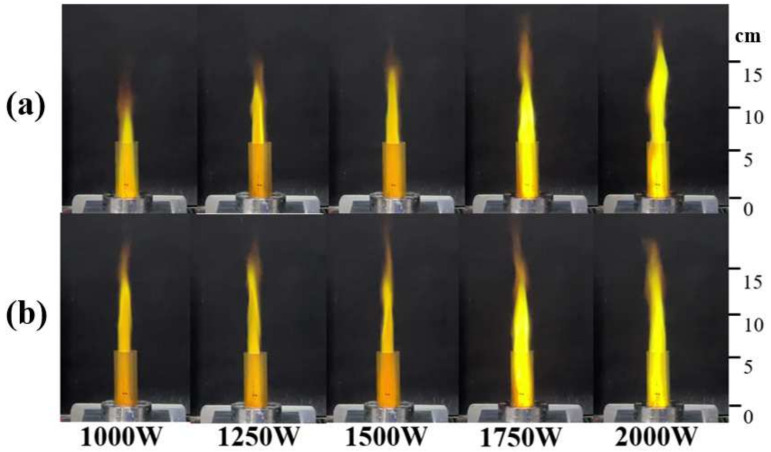
(**a**) Flame situation during initial combustion (5 s after the flame was observed); (**b**) Flame situation during steady combustion (25 s after the flame was observed).

**Figure 8 micromachines-13-00510-f008:**
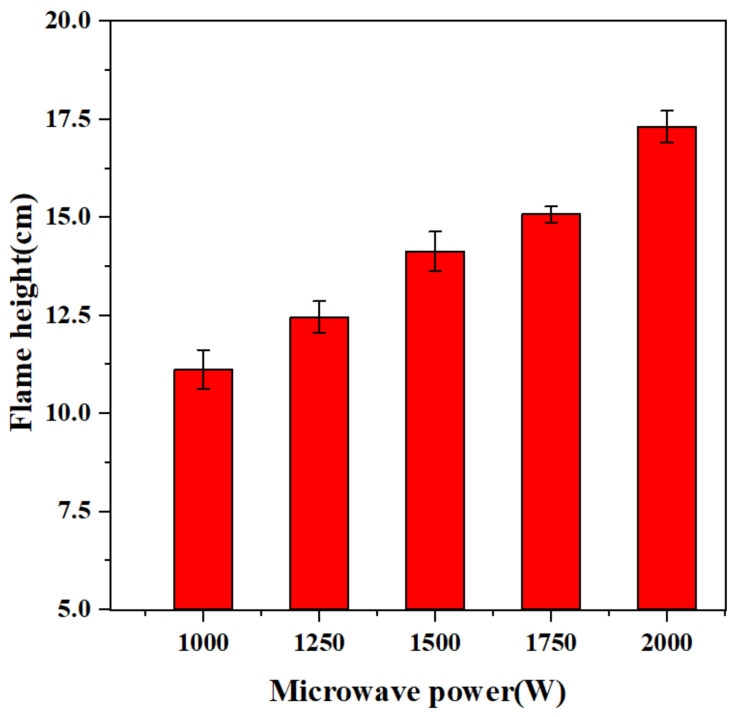
Effect of different input power on flame jet height.

**Figure 9 micromachines-13-00510-f009:**
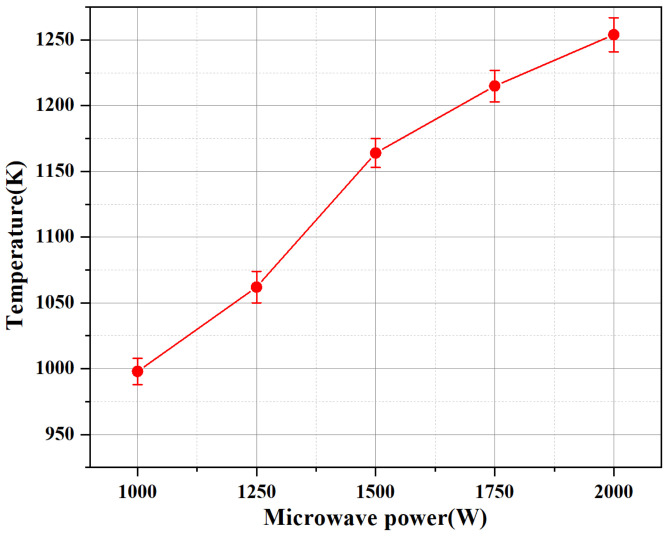
Effect of different input power on average flame temperature.

**Figure 10 micromachines-13-00510-f010:**
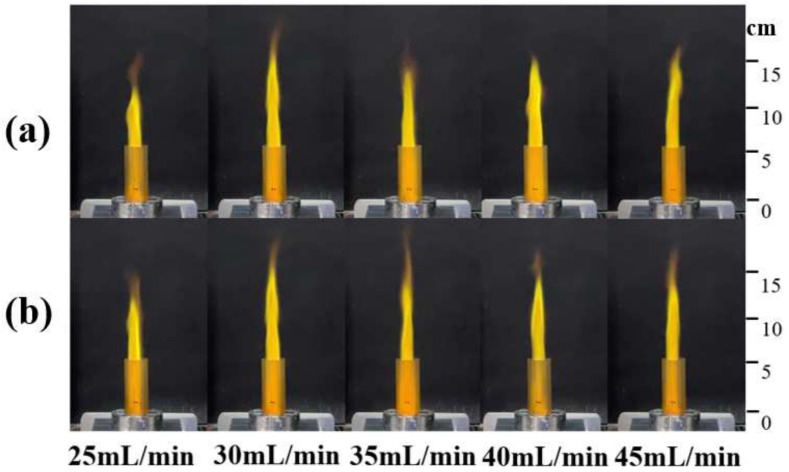
(**a**) Flame situation during initial combustion (5 s after the flame was observed); (**b**) Flame situation during steady combustion (25 s after the flame was observed).

**Figure 11 micromachines-13-00510-f011:**
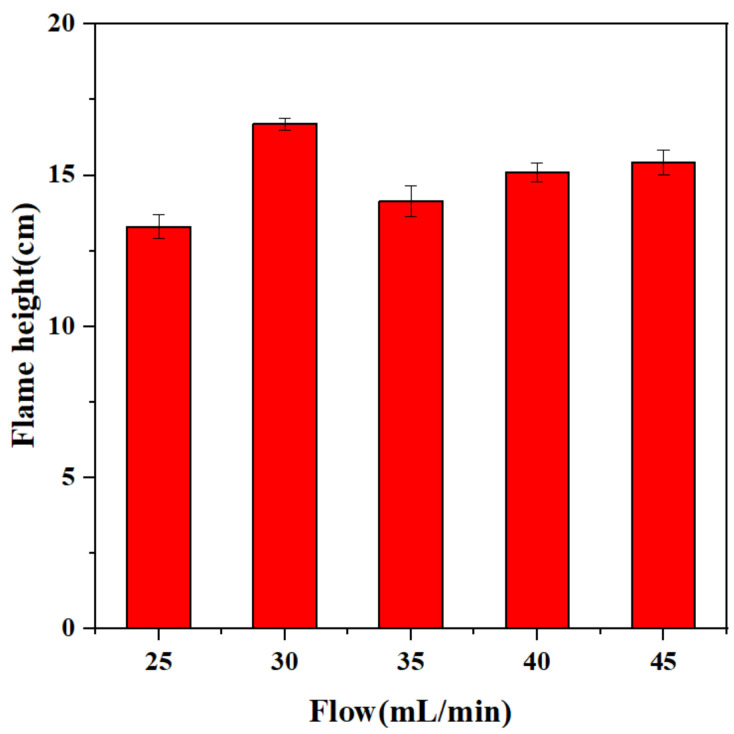
The effect of different propellant flow on the flame jet height.

**Figure 12 micromachines-13-00510-f012:**
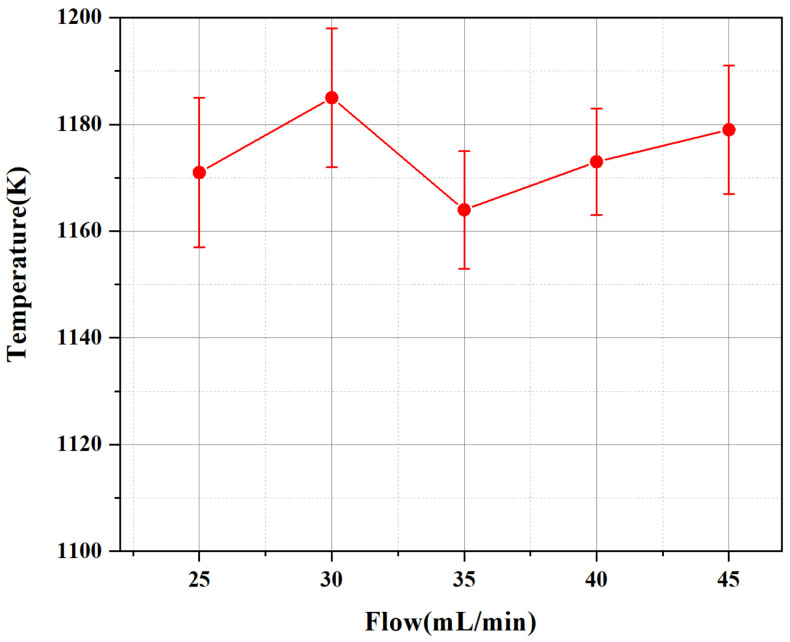
The effect of different propellant flow rates on the average flame temperature.

**Table 1 micromachines-13-00510-t001:** Dimensions of aluminum resonator.

Width (mm)	Height (mm)	Depth (mm)	Wall Thickness (mm)	Full Height Length (mm)	Half Height Section Length (mm)
90	47	200	2	100	70

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
