# Peer review of "Experimental Research on Microwave Ignition and Combustion Characteristics of ADN-Based Liquid Propellant"

_micromachines, 2022, doi:10.3390/mi13040510_

Round 1

Reviewer 1 Report

Please refer to the attached scanned copy for detailed comments

Author Response

Thank you very much for your  consideration and the  positive comments concerning our manuscript. I have deeply considered each comment given by reviewers, and admire those beneficial suggestions for a high scientific significance. I have made corresponding modifications in the revised manuscript according to the reviewer’s suggestions. The revised portions are marked in red in the revised manuscript, and the responses to the editor and reviewers’ comments are listed one by one in the following attachment.

Reviewer 2 Report

The paper describes the development of a new microwave igniter for the 
optimised use of ADN/air mixtures. The paper is very well written, but 
needs an editorial check of the English language and grammar. 
The paper shows nice results in the investigation of a novel microvwave 
resonator. However, there is no in-depth scientific discussion of the results. 
Hence, it can only be recommended for publication after major revision.

Generell comments:
When specifying a physical quantity, a space should be placed between the 
numerical value and the unit of measurement.

Introduction
In the paper, the authors present a microwave resonator to study the ignition 
and combustion of liquid ADN in air. Unfortunately, the important part of this 
plasma assisted ignition and combustion is far too short in the introduction. 
Although some references are given by [11-18], there is no discussion of the 
important physical and chemical processes, so that many questions remain 
unanswered. This part should therefore be dealt with in greater depth.
Just an example "chemically active particles" probably refers to the 
formation of atomic oxygen radicals. In addition to Ju's work (15.), 
I also refer to [1] and [2]. In these papers, the physical and chemical processes 
of PAI and PAC are explained very well.

The last paragraph gives the impression that only PAI is investigated with the 
help of a microwave resonator, but the influence of PAC is also investigated. 
This is also mentioned in the title.

Experimental setup
Air is both a carrier gas and, due to the oxygen, a reaction partner in 
combustion. Therefore, the term "auxiliary gas" is confusing here. 
How large was the air flow in L/min? 
According to Fig. 1, ADN was injected into the air flow before the ADN/air 
mixture enters the resonator. 
Are statements about the droplet size etc. possible here?
What is the flow velocity at the outlet?

In the introduction, the authors still speak of the importance of 
non-equilibrium processes and the formation of radicals as a result of microwaves. 
Here and later they talk about ionisation, i.e. the generation of radicals. 
Ions may also be formed here, but these are not decisive for ignition.

When microwave power is mentioned, does it always mean the power of the 
magnetron?
Is the micorwave power sufficient to form a microwave discharge or 
is the microwave field strength in the resonator
cavity was below that required to initiate or even sustain a
plasma without a flame? In Fig. 5 the electrid field strength is shown. 
Can the authors explain in section 2 also the stregth of the reduced electrical 
field?
Table 1 can be deleted. All details can be given in the text.

"In order to realize the ignition ... " to reach a specific electrid field 
strength a specific electrid field strength is necessary? This part has to be 
improved.
"Comsol Multiphysics" needs a reference.

p.4, l. 120: Figure 3 should be replaced with Figure 4. 
It would be helpful to show the coordinates and dimsensions in Fig. 4 and 5. 
Also the unit (V/m) should be included in Fig. 5.
Fig. 6 is nearly not explained. This is a result of the simulation?
Why 11 mm is chosen and not e.g.15 mm?

In section 2.3 again only microwave ignition is discussed. I would recommend to 
discuss here the outcome, if a prellant flo greater that 45 mL/min is used.

Results
The influence of microwave power on ignition is discussed very briefly in 
Fig. 7a. Fig. 7b is not explained. Here, too, the PAC part is neglected. 
What means "gas to be broke down"? 
In Fig. 7 (and Fig. 10) the unit mm should be included. Is this the coordinate 
of the 11 mm distance of the inner conductor?

Discussion by ig. 9 is not very successful. Is the temperature increase a 
consequence of the higher coupled energy (then the temperature would also have 
to increase only in air with increasing microwave power). Or is it a 
consequence of a stronger electric field and thus an increase in radicals, 
i.e. the amplification of the plasma-physical processes?

p. 9. l 212: "after 45 mL/min"? I recommend to incorporate this paragraph in 
subsection 2.3.

p. 10, l. 237: the height of "the plasma flame jet increased"  

[1]    S. Starikovskaia, Plasma assisted ignition and combustion, 
J. Phys. D: Appl. Phys. 39 (2006) R265-R299.
[2]    A.Y. Starikovskii, Plasma supported combustion, 
Proc. Combust. Inst. 30 (2005) 2405–2417.

Author Response

Thank you very much for your  consideration and the  positive comments concerning our manuscript. I have deeply considered each comment given by reviewers, and admire those beneficial suggestions for a high scientific significance. I have made corresponding modifications in the revised manuscript according to the reviewer’s suggestions. The revised portions are marked in red in the revised manuscript, and the responses to the  reviewers’ comments are listed one by one in the following attachment.

Round 2

Reviewer 1 Report

Many grammatical errors need to be fixed. The paper may be accepted but subjected to revision by a native English speaker for language. 

Reviewer 2 Report

All my criticisms of the original version have been satisfactorily incorporated. Therefore I can agree to a publication.